# Radar Interferometry as a Monitoring Tool for an Active Mining Area Using Sentinel-1 C-Band Data, Case Study of Riotinto Mine

Joaquin Escayo [1], Ignacio Marzan [2,*], David Martí [3], Fernando Tornos [1], Angelo Farci [4], Martin Schimmel [2], Ramon Carbonell [2] and José Fernández [1]

1 Instituto de Geociencias (IGEO), CSIC-UCM, Doctor Severo Ochoa, 7, 28040 Madrid, Spain; jescayo@ucm.es (J.E.); f.tornos@csic.es (F.T.); jft@mat.ucm.es (J.F.)
2 Geosciences Barcelona, GEO3BCN-CSIC, LLuis Sole Sabaris s/n, 08028 Barcelona, Spain; mschimmel@geo3bcn.csic.es (M.S.); ramon.carbonell@csic.es (R.C.)
3 Lithica SCCL, Av. Farners 16, 17430 Sta Coloma de Farners, Spain; dmarti@lithica.net
4 Atalaya Mining, La Dehesa s/n, 21660 Minas de Riotinto, Spain; angelo.farci@atalayamining.com
* Correspondence: i.marzan@csic.es; Tel.: +34-917-287-233

**Abstract:** Soil instability is a major hazard facing the mining industry in its role of supplying the indispensable mineral resources that our societal challenges require. Aiming to demonstrate the monitoring potential of radar satellites in the mining sector, we analyze the deformation field in the Riotinto mine, Spain. We propose a new method for combining ascending and descending results into a common dataset that provides better resolution. We project the LOS measurements resulting from both geometries to a common reference system without applying any type of geometric restriction. As a projection system, we use the vertical direction in flat areas and the slope in steep topographies. We then identify and remove outliers and artifacts from the joint dataset to finally obtain a deformation map that combines the two acquisition perspectives. The results in the Atalaya pit are consistent with GNSS measurements. The movements observed in the rock dumps were unknown before this study. We demonstrate the great potential of the Sentinel-1 satellite as a complementary tool for monitoring systems in mining environments and we call for its use to be standardized to guarantee a safe and sustainable supply of mineral resources necessary for a just technological transition.

**Keywords:** InSAR; Sentinel-1; Coherent Pixel Technique; ground stability; mine monitoring; sustainable mining

## 1. Introduction

The change in the energy model and the growing technology and environmental awareness are, currently, the main challenges facing society, all of which impose the need for a safe and sustainable supply of mineral resources, especially critical metals. Over the last few decades, several principles and strategies to promote sustainability through the whole mining cycle have been proposed by organizations such as the United Nations, the World Bank, the International Council on Mining and Metals (ICMM), and notably the European Union [1–5]. These actions are intended to establish a commitment to resource development in a socially and environmentally responsible manner. In addition to this, the prominence of the Social License to Operate (SLO) for any mining project can determine its viability. In this context, the mining industry must minimize its impact, improve transparency, and work to ensure the safe and sustainable supply of mineral resources that social challenges require.

Ground motion is one of the major hazards that the mining industry confronts. Earthworks, digging, and pumping affect ground mechanical integrity that may cause subsidence, landslides, and collapses; disasters that often have serious consequences, economically,

environmentally, and in terms of human life. Typically, ground stability monitoring uses ground-based instruments such as inclinometers, extensometers, and Global Navigation Satellite System (GNSS) receivers, which are of very high accuracy but provide discreet measurements as full coverage with 1D sensors is not feasible. By contrast, satellites allow extensive monitoring with each acquisition, covering areas where other sensors do not reach: intermediate and peripheral zones of a terrestrial network, zones of limited or dangerous access, or zones that were considered of low priority for monitoring. Furthermore, satellites are reliable, autonomous, non-invasive, and provide high-quality unbiased data repeatedly at low cost, all of which are ideal characteristics for a monitoring system. For these reasons, we have been working in recent years to promote the use of radar satellites, especially Sentinel-1, as a complementary tool to improve the monitoring capacity of mining stakeholders.

In orbit since 2014, Sentinel-1 has shaken up the Synthetic Aperture Radar (SAR) world with a wide range of capabilities, especially long interferometric series (InSAR). The Sentinel mission belongs to the Copernicus program, which is the result of a collaboration between the European Commission and the European Space Agency. Sentinel aims to provide free access to Earth's global coverage at a medium resolution and high revisit time, up to 6 days with a constellation of two satellites (Sentinel-1A and Sentinel-1B). It is worth mentioning that since December 2021 a technical anomaly is affecting the Sentinel-1B SAR electronic subsystem that is causing the interruption of data acquisition. At the time of writing this article, it is unknown if the satellite will be able to resume its operation plan. For this reason, an effort is underway to advance the launch of Sentinel-1C to the first half of 2023. Sentinel-1 carries on board a C-band radar sensor that, although not designed for high-resolution studies, we will see that it is a precious tool for continuous ground motion monitoring at a mining scale. Several studies show the ability of Sentinel-1 to effectively measure surface deformation at an engineering scale [6–8]. However, this technology is not normalized as a continuous ground instability monitoring tool due to the complex processing and high computational demands required. Our project InTarsis aims to bridge this gap and improve the monitoring capacity of the mining sector by standardizing the use of Sentinel-1 as a monitoring tool. Our strategy is to devise an algorithm that automates the processing of Sentinel-1 data, providing to the end-user a regular update of the surface deformation in the mining area. In this work, we describe the initial step of the InTarsis project, which is the characterization of the deformation field as the baseline for the monitoring of the Riotinto mining, Spain.

This paper is structured in two main parts. We begin with a historical review of field deformation in the area using all available data from the ESA radar catalog. Next, we conduct a comprehensive analysis focusing on the unstable zones using ascending and descending Sentinel-1 images. We combine them in a new post-processing method in which we exploit the information provided by each velocity measurement to take advantage of the outstanding resolution capacity of Sentinel-1. Sentinel-1 constellation offers a 6-day revisit time for its pre-defined acquisition mode for land and coastal zones (Interferometric Wide swath IW) with a 5 × 20 m ground resolution [9]. Combined with the free and almost instant availability of the data, as its part of the Copernicus observation program, makes Sentinel-1 a good option for monitoring applications [10–12] and especially in mining environments [13–19].

The InTarsis project is part of the RawMatCop program, which aims to develop skills, expertise, demonstrations, and applications of Copernicus data to the raw materials sector. It is co-funded by the European Commission (DG for Internal Market, Industry, Entrepreneurship, and SMEs) and the EIT RawMaterials (RM Academy). The EU aims to facilitate the exchange of best practices among its member states to improve the sustainable and safe supply of raw materials to the EU economy and society [20].

## 2. Riotinto Mine Context

The Iberian Pyrite Belt (IPB) is a large Variscan metallogenic province around 250 km long and 20–70 km wide along Portugal and Spain, hosting the largest concentration of massive sulfide deposits worldwide [21,22]. The main mining interest is and has been precious metals (Au, Ag), base metals (Cu, Zn, Pb), and sulfuric acid. The major mineralization, as volcanogenic massive sulfides and related stockworks, was deposited in pull-apart intracontinental marine basins related to the closure of the Rheic Ocean due to the collision between the South Portuguese Zone, an allochthonous terrane, and the autochthonous Iberian terrane during the late Devonian-Visean [23–25].

The Riotinto mining district is the largest of the nine giant massive sulfide deposits of the Iberian Pyrite Belt (IPB) and one of the largest sulfur anomalies in the Earth's crust. The dimensions of the orebody and its intense (and sometimes turbulent) long history make it a mining emblem. Since 4000 BC, mining activity is documented by Tartessians, Phoenicians, Romans, Arabs, Spanish, British, and modern multinational companies [26]. Currently, the Atalaya Mining company owns the mine and focuses on copper production. The mine area covers ~36 km$^2$ divided into two zones, the Atalaya abandoned open pit, and the Cerro Colorado active pit (Figure 1). The geology of the area is quite complex and includes a thick volcanic unit, the Volcanosedimentary Complex [27] with basaltic flows and sills intruding shale overlain by several superimposed rhyodacite domes. These rocks are covered, through an out-of-sequence large thrust, by a synorogenic flysch-like sequence, the Culm Group. These volcanic rocks have been affected by the aforementioned Variscan orogeny causing widespread folding and thrusting structures, which were later reactivated during the Alpine orogeny [27]. This deformation has produced a complex network of fractures that nowadays controls the hydrology and the geotechnical behavior of the mined rocks.

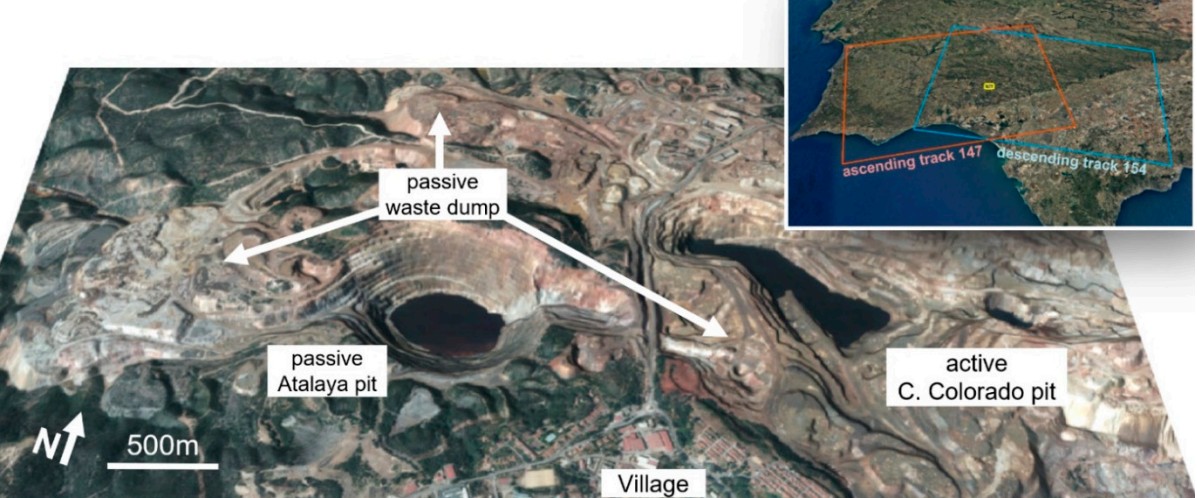

**Figure 1.** Main mining elements in Riotinto mine. The mining site is divided into one active zone to the east and one passive zone to the west. The small map shows the localization of the Riotinto's mine (yellow square) and the Sentinel-1 footprints, ascending (red) and descending (blue), of the selected tracks to be used in Section 4. In this specific configuration, both incidence angles on Riotinto are close to 42°.

Before 1907, the mining activity was underground, as a result, countless galleries crosscut the site. Then, after major collapses, underground works were replaced by several open pits, being most of them nowadays infilled. At its peak, the Atalaya open pit was the largest in Europe, being finally abandoned in 1994. The dimensions and elliptical shape (1200 m long, 900 m wide, and 350 m deep) make it a reference in open mining. However, the interplay of the previous structures has produced the widespread collapse of

old underground works and the sliding of faces within the pit. Of special interest is the east side of the open pit where the conjunction of shale and a major fault weakens the wall, showing a continuous sliding process linked to the old mining activity. The concern caused by the deformation observed in this area is the main motivation behind this study, and it will be discussed in detail in Section 4. The Atalaya pit is surrounded by passive dumps, of around 3.5 km$^2$, which we will also analyze.

Nearby, to the east, the Cerro Colorado pit is currently the active part of the license. It is a complex mine with 3 pit faces, resulting in an irregular shape, with a maximum length of 2020 m, a maximum width of 850 m, and a maximum depth of 230 m. In 1964 began the digging works, but due to the low profit, the activity stopped in 2001. As a consequence of the improvement of the metal market, Atalaya Mining resumed its mining activity in 2015, and it is now at full capacity. According to the estimated reserves, it can extend to 17 or 25 more years.

The mining license is surrounded by 3 small towns, El Campillo, Nerva y Minas de Riotinto. Those are deeply connected to the mining history as noted for its Roman and pre-Roman archeological remains. After a 14-year halt, the mining activity has revitalized the commercial activity in the zone. Beyond this positive impact on the community, the mining company works to reduce any negative mining impact, especially regarding the environmental liabilities under its responsibility. This work focuses on the stability evaluation of the passive mining elements, which mainly are the Atalaya open-pit and the surrounding rock dumps.

## 3. InSAR Historical Review (ERS, ENVISAT, Sentinel-1)

The main objective of the InTarsis project is to devise a ground stability monitoring system based on InSAR to improve Atalaya Mining's control over its environmental liabilities. The first step consists of characterizing the deformation field in the target area to establish the monitoring baseline. Therefore, we have conducted a historical review of the SAR data in the ESA archives to identify the main deformation patterns. We downloaded all available data for the study area from the ERS (1991–2003), ENVISAT (2002–2012), and Sentinel-1 (2014—present) constellations. All three ESA constellations follow the same strategy for continuous monitoring of the planet with global coverage at medium resolution using C-band sensors suitable for global mapping and monitoring of areas with low to moderate penetration.

We analyzed 481 images from 1992 to 2017 with different acquisition geometries, ascending and descending, from all tracks and frames covering the area. Finally, for this review, and depending on availability and suitability, we have selected 3 datasets:

- 39 images from ERS1 and ERS2, descending track-366 from 1992 to 2001.
- 22 images from ENVISAT, descending track-366 from 2003 to 2010.
- 33 images from Sentinel-1, descending track-154 from 2015 to 2017.

The main reason for working with descending images at this step is that it is the most suitable geometry to study the affected zone in the Atalaya open-pit, which is the main target of this study. For processing these datasets, we applied the Advanced Differential Interferometric SAR technique (A-DInSAR) using the SUBSIDENCE-GUI interferometric software based on the Coherent Pixels Technique (CPT) (Figure 2) developed at the Polytechnic University of Catalonia (UPC) and DARES [10,28–30]. This software allows the user to perform the full interferometric chain by generating the interferograms starting from the SLC product and calculating the time series by applying different approaches based on multilooked images (coherence-based) and full-resolution techniques such as the recently developed Temporal Phase Coherence [31].

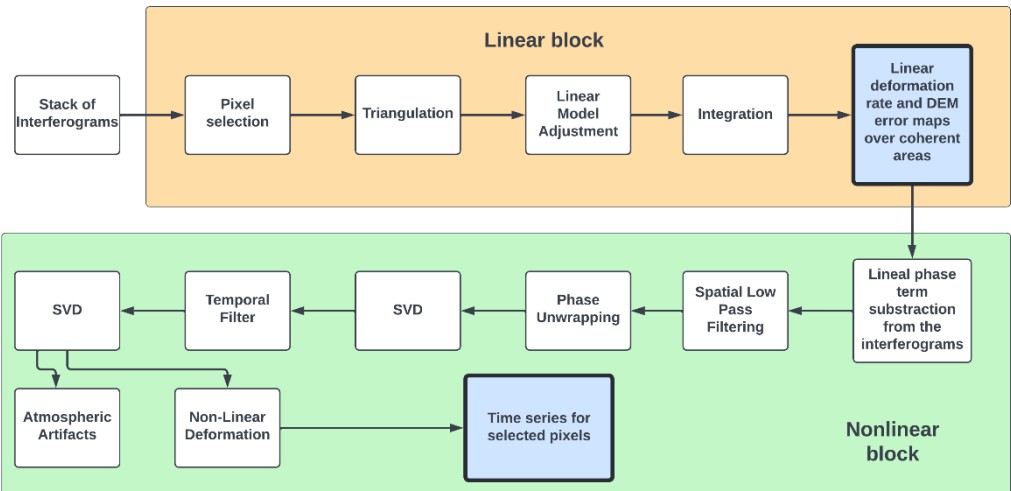

**Figure 2.** Workflow of the CPT algorithm for time series calculation. One of the main differences between the CPT technique with other coherence-based techniques is the initial estimation of the linear term of the displacement (Linear block).

We clipped a region of interest (ROI) of 10 × 11 km around the mining zone and we selected a multilooking window of 3 × 15 samples which results in a ground resolution of around 60 × 60 m for the three satellite constellations. The processing parameters generated a network of interferograms for each constellation that are worth comparing (Figure 3). The interferogram network of ERS satellites covers 5 years of images from mid-1995 to the end of 2000, images producing very low-quality interferograms are discarded. The resulting number of images available is not very high, around 4 images per year with high temporal baselines (temporal difference between acquisitions) which causes temporal decorrelation. Besides this, the spatial baseline (the spatial distance between the two sensor positions when the SAR pairs are acquired) is very large and this will affect the quality of the interferograms and the measurement coverage. The Envisat's network of interferograms shows less dispersion in the spatial baseline, but very important gaps in the temporal baseline, especially between 2007 and 2010, causing loss of coherence and therefore big zones without results, especially in areas where changes in the surface have been more important at that period. And finally, the Sentinel-1 cloud for the selected period, 2015–2017, shows a spectacular improvement in both temporal and spatial baselines, which will provide excellent interferogram quality and measurement coverage. The narrow Sentinel-1 orbital tube makes the spatial baselines very small, between ±100 m, which ultimately results in very consistent displacement measures [32]. This dataset Sentinel-1 compiles images every 24 days, which we think is a good comprise for this study, even though the revisit time is of 6 days since April 2016.

The processing results of the three constellations are shown in Figure 4. It is remarkable the spectacular improvement in coverage from ERS to Sentinel-1. Within the ROI, we could determine 3586 velocity values for the ERS constellation, 5507 values for ENVISAT, and 14,091 values for Sentinel-1. In the case of ERS and ENVISAT, the gap in data coverage could be explained by the large baselines. When the temporal or spatial baselines between two images are too far apart, signal decorrelation may be locally high, and the results from these areas are eliminated during the processing by low coherence filtering. The poor coverage of the ERS dataset is especially remarkable, preventing the identification of any deformation pattern in the scene. The ENVISAT dataset shows better coverage, and three zones of deformation in tailing zones can be identified. Finally, the Sentinel-1 dataset shows dense coverage along the scene providing information from almost all non-dense vegetated areas and water bodies. The very small Sentinel-1 baselines preserve the coherence between images and provide a spectacular improvement in measurement coverage. The deformation areas unveiled by ENVISAT are now well mapped in extension and value. Besides this, it

uncovered a new deformation zone, indeed, the most conspicuous deformation zone in the area of the east wall of the Atalaya pit (Zone 4). This area is being monitored by the mining company and will be discussed in detail in the following sections.

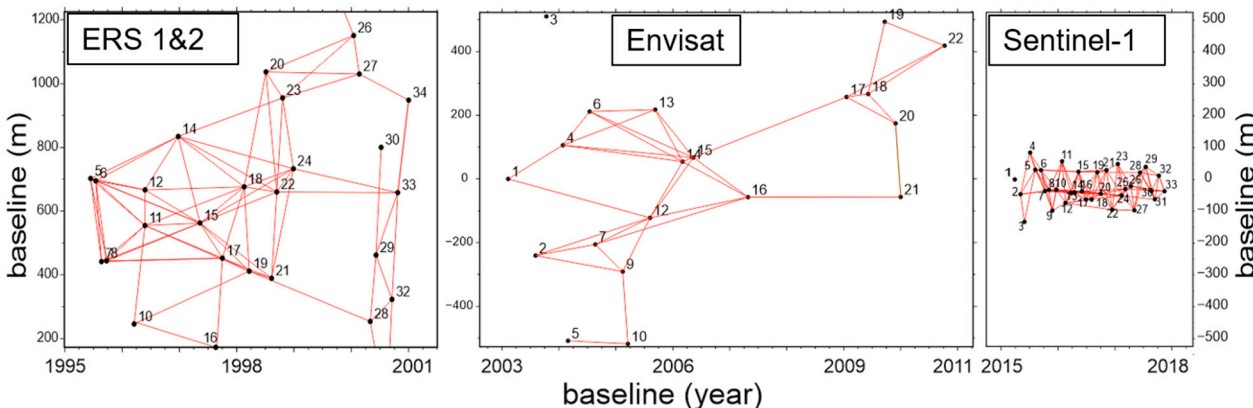

**Figure 3.** Interferometric cloud of the three constellations ERS, Envisat, Sentinel-1. The numbers in the graphs correspond to the radar images (numbered by its acquisition date) and the connection lines to the interferometric pair. The temporal and spatial baseline improvement provided by Sentinel-1 is remarkable.

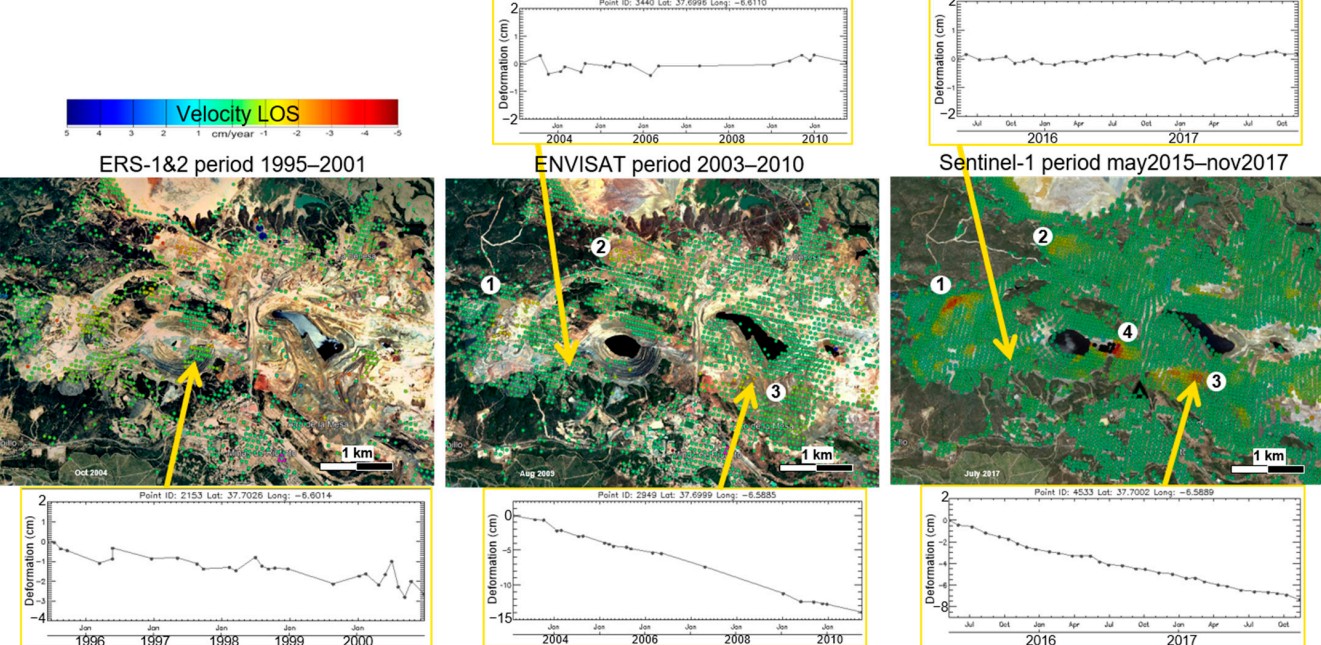

**Figure 4.** Historical review of Riotinto mine with 3 ESA satellites: ERS-1&2, ENVISAT, and Sentinel-1. Result for descending orbits. Dots represent mean velocity values (cm/y, in line of sight direction, LOS) of each cell. Green dots mean stable and reddish colors mean movement away from the satellite. The numbers on the maps show four areas under deformation discussed in the following sections. The graphics show accumulated deformation in stable and unstable points. It is worth noting the coverage improvement with Sentinel-1.

## 4. Combining Ascending and Descending Sentinel-1 Images to Analyze Unstable Areas

The 4 zones showing displacement (Figure 4) are located in the inactive part of the mine, the Atalaya pit, and its surroundings. We focus on these passive elements because, ideally, they should remain stable and, in case there is movement it should be under control

to prevent risks. We will focus the analysis on this area using both ascending and descending Sentinel-1 images for mapping and quantifying the displacements and the potential causes. We use the aforementioned results from the descending track 154 and, in parallel, we processed the ascending dataset corresponding to track 147, using the same software, the same period, and similar processing parameters (Figure 1). The challenge is now to combine both processing results, descending and ascending, to produce a comprehensive description of the deformation field. Due to their opposite perspective, both processing results identify the same anomalies but provide complementary information about the scene. Geometric distortions and shadows are complementary, ascending dataset performs better in characterizing eastern slopes, and the descending dataset western slopes (Figure 5). It is interesting to observe how ascending provides little information from Zone 4 (the eastern steep wall of Atalaya with western dip), but, on the contrary, it provides better resolution of Zone 3 which is bounded by slopes facing east. We assume that velocity values between −0.5 and +0.5 cm/y are below our resolution capacity so those are stable points.

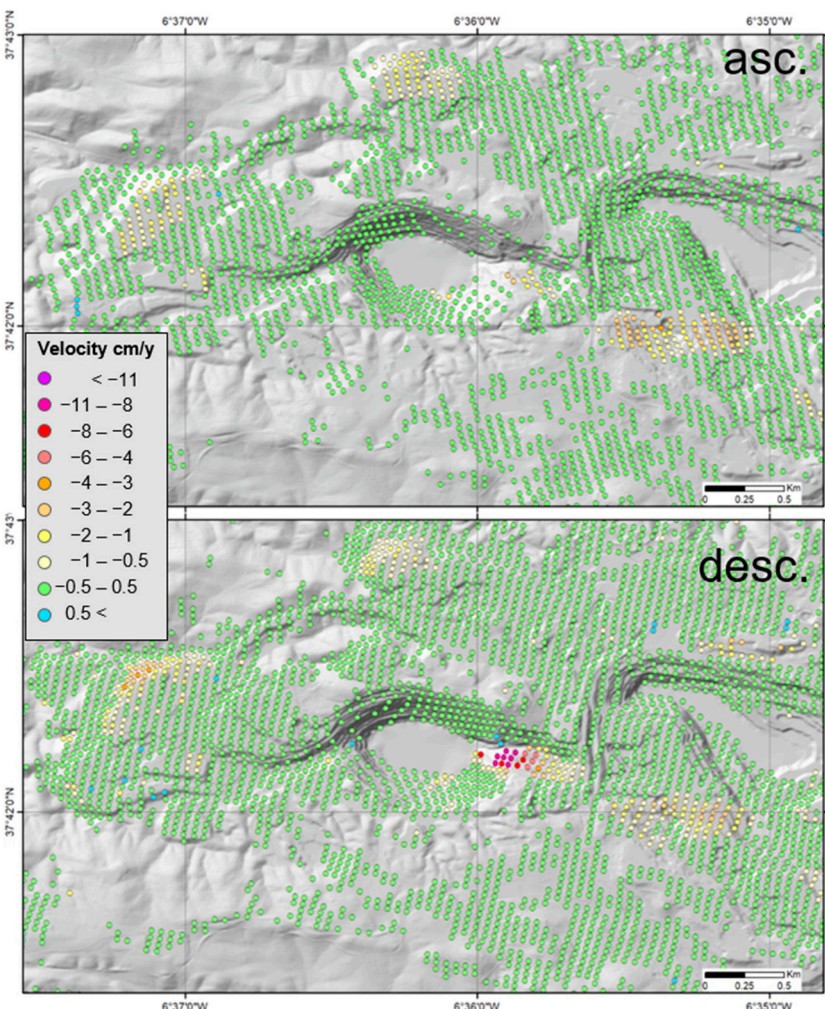

**Figure 5.** Map of displacement measured in LOS. Negative values mean moving away from the satellite. Ascending geometry up, descending geometry down.

Ascending and descending displacement results are in different projections; therefore, the integration requires a reprojection to a common reference system. The velocity vector of a surface displacement calculated from the InSAR processing is only its projection to the LOS direction (the satellite Line of Sight). As a consequence, we only observe if the target is moving towards the satellite (+LOS, positive by convention) or moving away from the satellite (-LOS). The velocity in the LOS component (Vlos) is a great tool for

mapping areas in motion and preliminary evaluation of the deformation field. However, for accurate quantification, Vlos needs to be projected into the georeferencing system. On the other hand, the decomposition of the Vlos into the geocoordinates xyz entails a problem inherent to the image acquisition: radar satellites have a low sensitivity for displacements in the track direction, $\pm 10°$ north in the case of Sentinel-1. A common strategy for Vlos decomposition is considering only cases where the north-south component of displacements can be neglected; then, Veast-west and Vz can be estimated wherever ascending and descending measurements are available [33–36]. Another projection strategy applied to the study of landslides assumes that the movement occurs mainly in the direction of the maximum slope [37–40]. In this work, we follow a similar strategy assuming Vlos is the slant-range component of Vslope (velocity along the maximum slope direction) in the steep topographies. With regard to flat topographies, we assume displacements are entirely in the vertical direction, so the Vlos in this case is the slant-range component of Vz (vertical subsidence). It is worth pointing out that these projections can have an amplification effect that can be very important, providing meaningless velocity values on some unfavorable topographies. That is why it is essential to collect any additional information about the target from the ground level: terrestrial velocity measurements, mechanical behavior of the terrain, type of sliding; as well as any detail that can help to interpret the results.

### 4.1. Projection to a Common Reference System

The target zone is located in a stable Paleozoic basement of gentle mountains that characterize a rounded shape landscape. In regard to the potential causes of ground motion, it is worth saying that there are no unconsolidated sand-gravel aquifers, there are no volcanoes, and the tectonic activity is irrelevant in this geological context [41]. Therefore, it is reasonable to assume that the observed displacements in the abandoned elements of the mine, pit, and dumps, are caused by the loss of mechanical integrity. As mentioned earlier, we assume that displacements are exclusively in two directions: (1) Vz (vertical displacement) in zones of a gentle slope, and (2) Vslope (along the slope) in steep zones. The threshold slope value between those two trends is deduced from the representation of slopes vs. Vlos (Figure 6). This graph shows that the largest displacements occur for slopes greater than 13°, as a consequence, we assume that 13° is the slope threshold between a Vz dominance and a Vslope dominance.

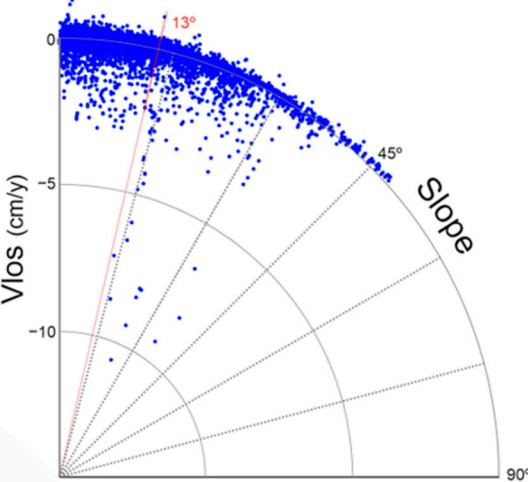

**Figure 6.** Cross plot of slopes vs. Vlos. The red line shows the 13° threshold from which the largest displacements begin.

The implementation of this strategy can be reduced to solve a simple problem of geometry where the projection of the velocity vector can be found by dividing Vlos by the cosine of the angle σ (cos σ) formed with the slope in every point. In the flat areas, we

project to the vertical, which is equivalent to assuming a 90° slope. Therefore, to get $\cos \sigma$ we define the unit vector uSLOPE in the slope direction, and aLOS (ascending) and dLOS (descending) in the LOS direction. The uSLOPE is defined by the angles S (slope) and A (aspect) (Figure 7a). The aLOS and dLOS are defined by the incidence angle of the beam, θ, and the azimuth angle of the LOS direction, α (Figure 7b). In our test-site of Riotinto, with the selected ascending and descending tracks, it turns out to have a very similar incidence angle on ascending and descending geometries, $\theta \approx 42°$ (Figure 1). Besides this, Sentinel-1 LOS azimuth angle, α, is always 80° for ascending and −80° for descending.

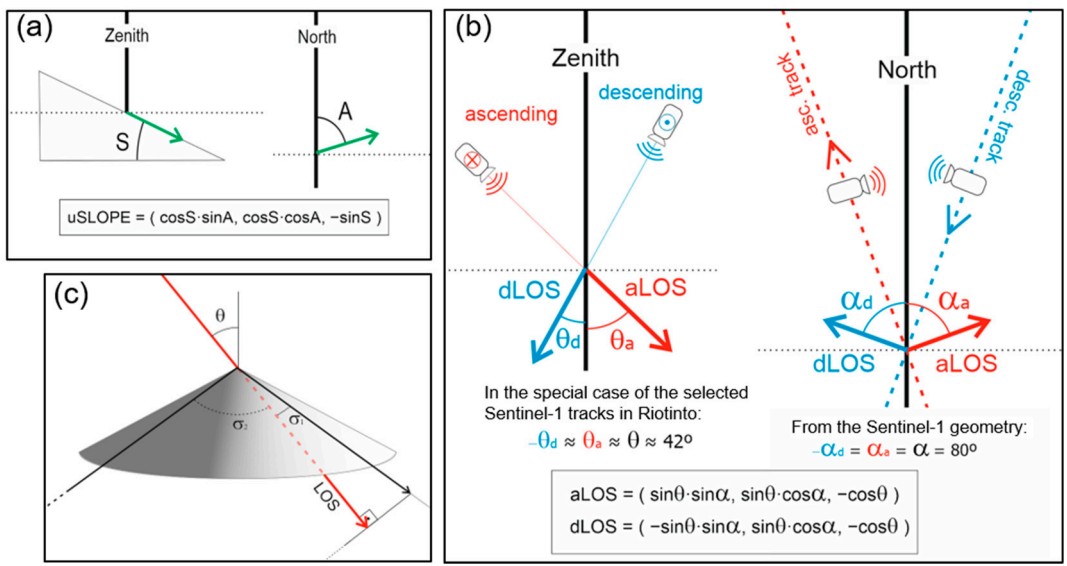

**Figure 7.** (**a**) Geodetic components of the slope unit vector in slope, uSLOPE. (**b**) Geometry of the line of sight (LOS) unit vector and geodetic components ascending, aLOS, and descending, dLOS. (**c**) Cone of all possible slopes and two angles (σ) with a LOS vector. Projecting LOS to the slope goes to extremely high values when σ approaches 90° (σ2).

The cosine of the angle (σ) between SLOPE and LOS is the result of the scalar product of their unit vectors:

$$\text{uSLOPE·uLOS} = \cos \sigma \tag{1}$$

The geodetic components are:

$$\text{uSLOPE} = (\cos S \cdot \sin A, \ \cos S \cdot \cos A, \ -\sin S) \tag{2}$$

where S and A are the slope and aspect angles, respectively. Whereas the uLOS decomposition depends on the acquisition geometry:

$$\text{uLOS} = \text{aLOS} = (\sin \theta_a \cdot \sin \alpha_a, \sin \theta_a \cdot \cos \alpha_a, -\cos \theta_a) \tag{3}$$

$$\text{uLOS} = \text{dLOS} = (-\sin \theta_d \cdot \sin \alpha_d, \sin \theta_d \cdot \cos \alpha_d, -\cos \theta_d) \tag{4}$$

Equation (3) is for ascending geometry and Equation (4) is for descending. In the special case of the selected Sentinel-1 tracks in Riotinto (Figure 7b, the incidence angle of the beam and the LOS direction meet:

$$\theta_a = -\theta_a = 42° \qquad \text{and} \qquad \alpha_d = -\alpha_d = 80°$$

Finally, Vslope can be calculated as:

$$\text{Vslope} = \text{Vlos} / \cos \sigma \tag{5}$$

However, when σ approaches 90° the cosσ approaches 0 and it is evident that LOS projection can produce extreme values that must be considered artifacts (Figure 7c). In general, this configuration is unlikely among velocity obtained from DInSAR processing since unfavorable topographies tend to produce geometric distortions, and therefore low coherence pixels that are normally removed during processing. However, in special locations with a peculiar topography, extreme Vslope values can be obtained, and the results must be interpreted accordingly. That is why it is important to use Digital Elevation Models (DEM) with a similar resolution to the radar image to reduce noise in the post-processing calculations. In our processing, we used a multi-looking parametrization of $3 \times 15$ getting a radar-range resolution of $60 \times 60$ m that can be translated to a flat Earth resolution of $40 \times 40$ m (using $\theta = 42°$). Hence, we resample the DEM using a cubic convolution algorithm to decrease the resolution to $60 \times 60$ m, filtering in this way the high topographic wavelengths that can produce that kind of artifact.

Finally, we projected the Vlos values, ascending and descending, to a common reference system: the local slope when this is greater than 13°, and the vertical if it is smaller (stable points are not projected). However, in the displacement affecting Zone 4 we had additional information from the local GNSS network that provided the motion direction that we used as projection (more detail in Section 5).

### 4.2. Integration of Ascending and Descending Measurements

Having ascending and descending velocity vectors in the same geodetic projection the following step is intended to combine them in a common dataset. The high precision of the Sentinel-1 orbit together with the high resolution of the sensor and the high density of the measurements make us consider a priori all the Vlos values equally valuable, including unfavorable geometries. Therefore, because of the scale and precision of the case study, our strategy is to preserve and exploit as many measurements as possible until the end of the post-processing. We start by considering all calculated velocities equally valid, and then we apply the following 3 steps:

1. Once ascending and descending datasets are in the same projection, we merged them into a common dataset of 8824 values (Figure 8). Then, ascending and descending can be jointly interpreted. However, the new dataset contains artifacts and outliers that need to be inspected.
2. Among the values projected to the slope (Vlos), those with negative cosσ and Vlos have been removed, 25 of them, because this combination causes artifacts (Figure 8). Values of cosσ < 0 entail LOS-slope angles greater than 90°, with this geometry, only positive Vlos (displacements approaching the satellite) can be projected downslope direction. On the contrary, the negative Vlos can be only projected in an upslope direction, that is why we call climbers to these artifacts produced by unfavorable geometries. We draw attention to the fact that by convention displacements approaching the satellite are considered positive.
3. We also removed isolated outliers, 61 of them, mainly located in the active part of the mine. The outliers have been mapped and removed when they conflict with neighboring values (Figures 8 and 9).

In the end, we got a final clean dataset of geo-projected velocities that combine both ascending and descending values (Figure 8). The proposed integration strategy has the advantage of preserving ascending and descending DInSAR processing results to the maximum. There are no a priori geometric restrictions. Then, artifacts (climbers and outliers) are removed from the final dataset. On the other hand, the downside is that some low-quality data may have escaped and integrated into the final dataset.

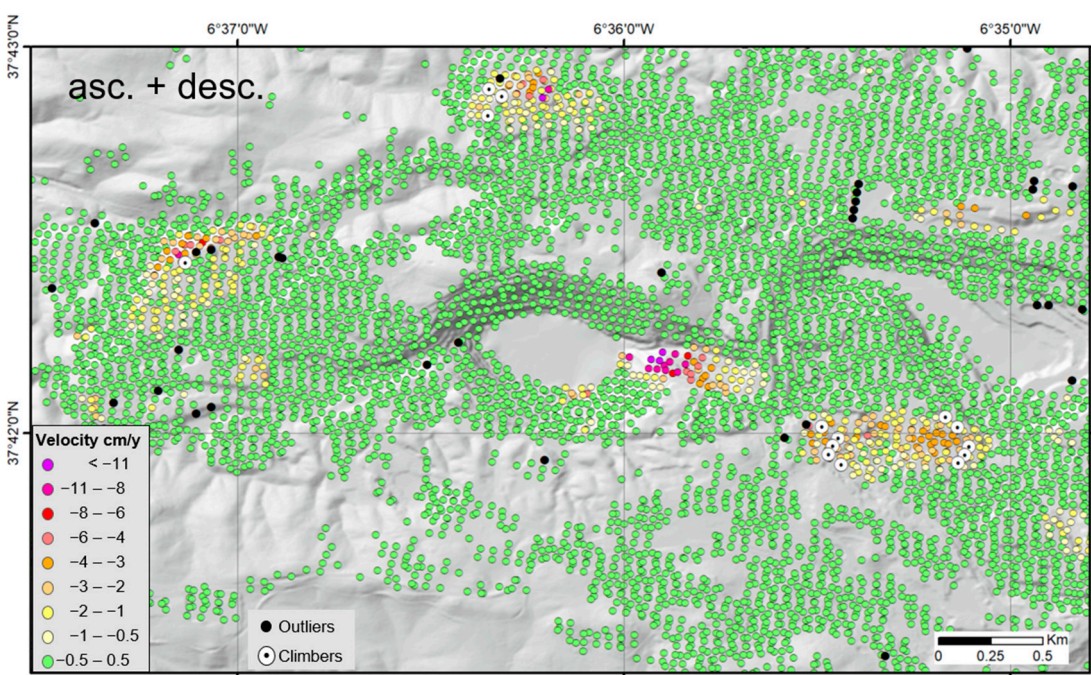

**Figure 8.** Displacement map integrating ascending and descending values. Velocity values quantify vertical displacement for slopes less than 13°, and displacement along the slope for greater slopes. Negative values mean downwards. Outsider values are in black and "climber" artifacts are the "vector out" symbols.

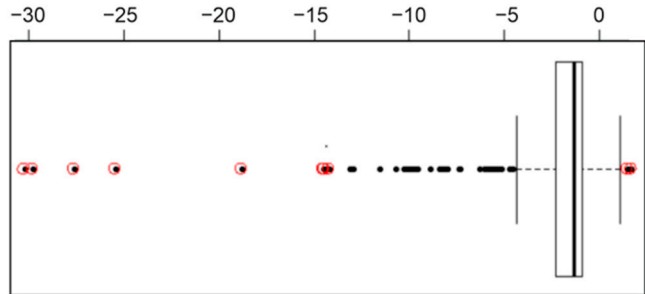

**Figure 9.** Boxplot of the joint dataset. Red circles show the outliers that have been removed because they conflict with neighboring values.

## 5. Results and Discussion

By interpolating the final dataset, we mapped the surface deformation field of Riotinto for the period 2015–2017 (Figure 10). It shows the aforementioned 4 anomalies (other anomalies linked to the active Cerro Colorado pit are not analyzed in this work). Displacements in zones 1 and 2 happen in rock dumps and were not known before this study. Values go from less than 1 cm/y for most of the extension, up to 5 cm/y close to some steep slopes. These dumps are made up of disaggregated materials and therefore their mechanical behavior tends to plasticity (Figure 11), which could justify differences between vertical subsidence in the flat summit and along the slopes in steep zones that we interpret as debris slides [42]. It is important to take into account that some displacements in Zone 2 are in the northbound slope, a direction of low sensitivity for the DInSAR technique since it is oriented N-S.

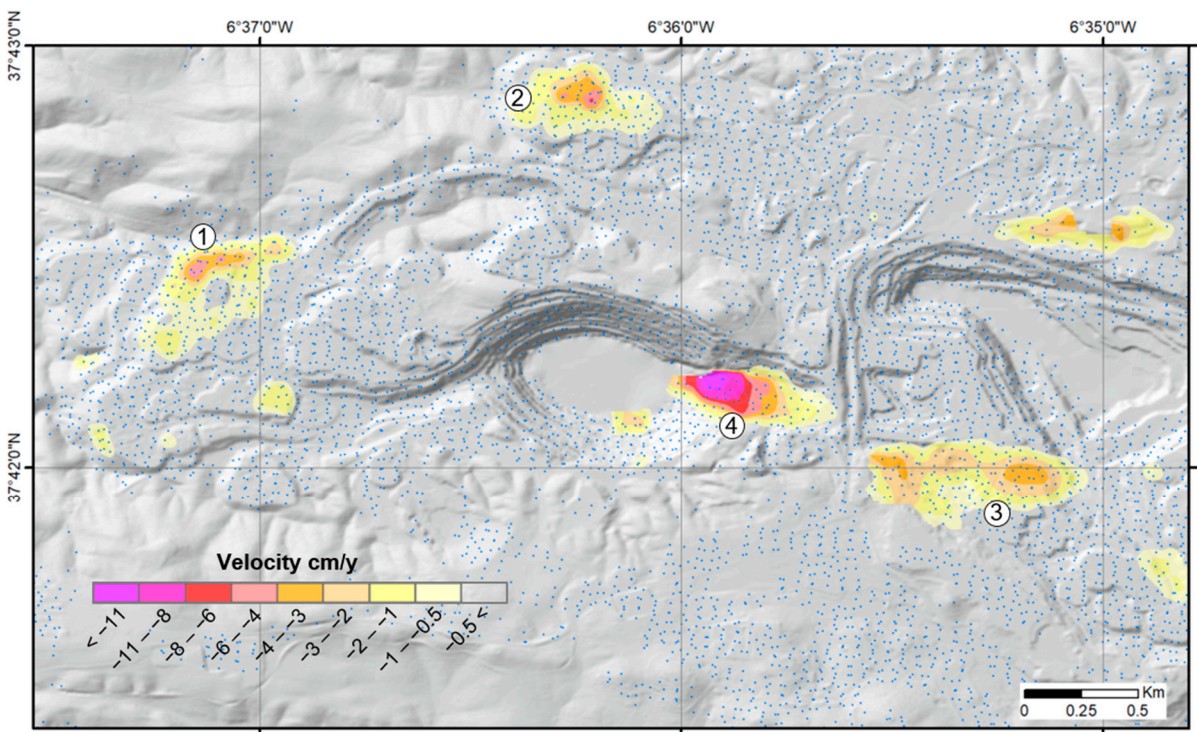

**Figure 10.** Deformation map showing the velocity anomalies interpolated from the displacement map, blue dots (Figure 8). Negative displacements are downward in flat zones and along the maximum slope in steep zones. The numbers show the four zones under discussion.

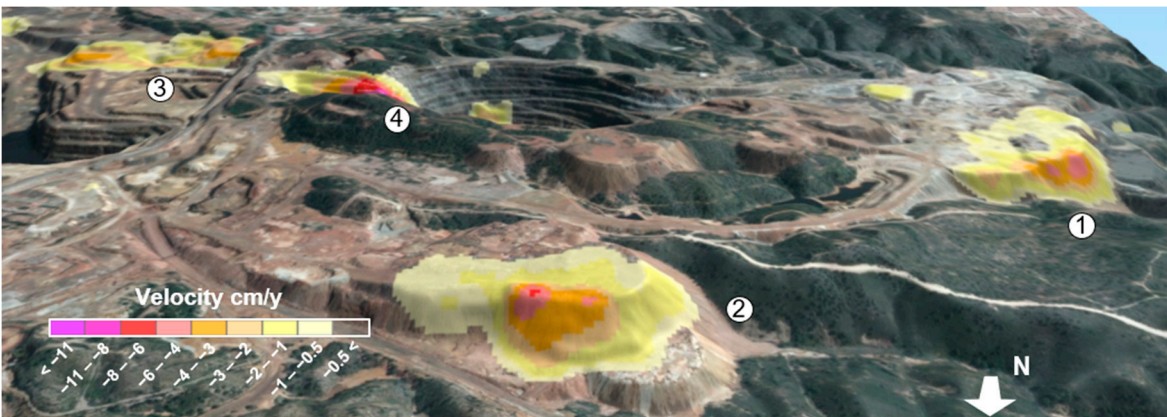

**Figure 11.** North-South view of the passive mining elements. The numbers are the four zones under discussion.

Despite good coverage and consistent data, ground-level data is required to confirm these results. Nevertheless, we suggest increasing the control of these two passive dumps. On the other hand, Zone 3 shows a pattern of displacement consistent with subsidence and collapses observed at ground level. Atalaya Mining monitors terrain instability in this area to prevent any risk. Zone 3 is an old ore deposit first mined through galleries and then open pit, and finally filled with mining waste and abandoned. It is believed that the cause of the ground instability is failures in the abandoned galleries. Furthermore, this area is affected by the mining plan in Cerro Colorado, changing its status to an active zone.

*Atalaya Open-Pit, Zone 4*

Thanks to the terrestrial monitoring conducted by the mining company at the east of the Atalaya open pit, we have been able to make a detailed analysis of the observed displacements. As previously stated, in Zone 4 the Vlos vector has been projected not along the maximum slope but to the main displacement direction determined by the local GNSS network (Figure 12). The local topography in the east wall of the open pit is a narrow convergent slope crossed by a fault zone and the shale of the Volcano-sedimentary Complex and the Culm Group [27]. This conjunction weakens the wall structure and breaks it into blocks (Figure 12). We interpret the velocity values observed as block slide type displacement inside this weakened zone [42]. From the average of the displacement vectors measured using GNSS, we determined a direction with angles of 42° dip and 331° azimuth that we used to project Vlos in this area. Assuming that all points on the blocks are moving in the same sliding direction, we compared in a graph four coinciding GNSS and Sentinel-1 measurements (Figure 12). The correlation is high, getting velocities values up to 11 cm/y. The small correlation mismatches could be explained by scale differences, while GNSS measurements are at specific point locations, InSAR measurements represent pixel averages (~40 × 40 m in flat Earth resolution). It is also worth saying that the GNSS survey markers are not cemented pillars but Feno-type nails, the integrity of which is more vulnerable to unexpected damages.

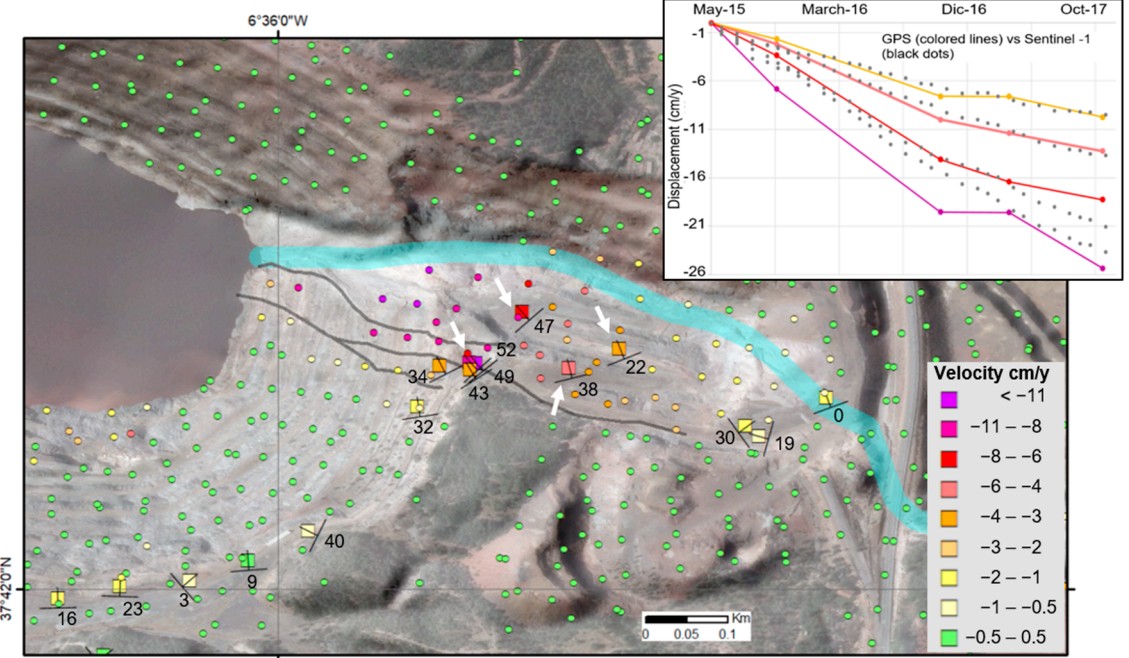

**Figure 12.** Zoom in Zone 4, convergent slope in the east wall of Atalaya pit. The block slide seems to be limited by the fracture zone (black lines) and the contact of the massive sulfides with the siliciclastic sediments of the Culm Group (thick blue line). The GNSS measurements are represented by colored squares. GNSS displacement vectors are marked with dip symbols (azimuth direction), and numbers (tilt degree). The circles are the calculated velocities from Sentinel-1 (Figure 8). The color scale is valid for both squares and circles. The white arrows show the GNSS—Sentinel-1 correlation points. The graph compares cumulative displacement from May 2015 to November 2017 between the Sentinel-1 (black dots) and GNSS (colored lines) displacement.

Taking into consideration our resolution capacity, we observe that the greater displacement in zone 4 affects only a corridor limited by the fault zone and the shale of the Volcano-sedimentary Complex and the Culm Group (Figure 13). No other relevant movements are observed in the pit apart from minor displacements in the south wall coinciding

with terraces that collapsed before the surveying period, which we interpret as debris slides. It is remarkable how Sentinel-1 is capable of measuring such small displacements even though they occur towards the north, which is a direction of low sensitivity for the sensor. The anomaly in the base of the wall covers the whole debris zone (~120 m) and shows displacements up to 3 cm/y. The anomaly in the upper wall is smaller (~50 m) and with values less than 1 cm/y, in good agreement with the GNSS measurements in this zone.

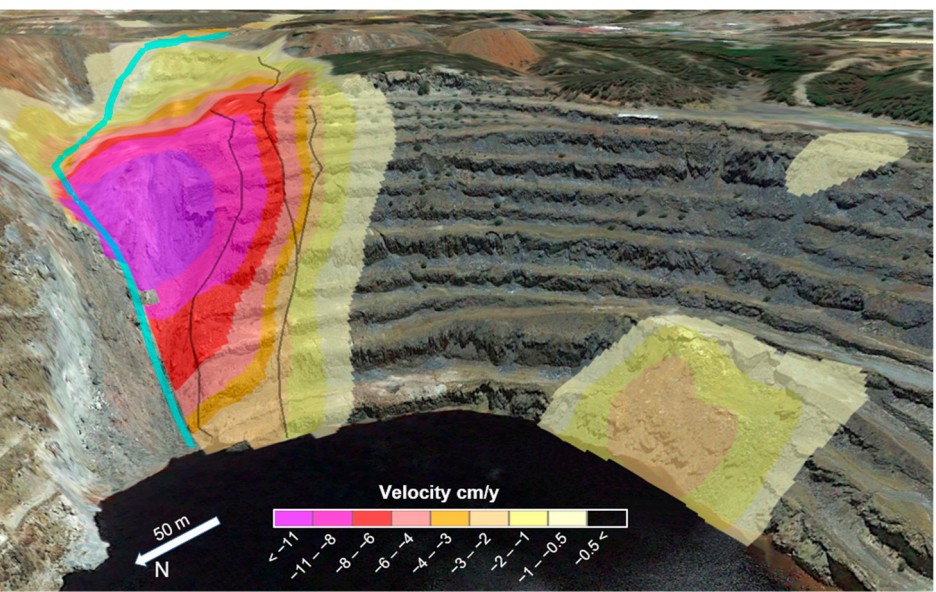

**Figure 13.** Zoom in the surface deformation map on the Atalaya pit. The block slide seems to be limited by the fracture zone (black lines) and the contact of the massive sulfides with the siliciclastic sediments of the Culm Group (thick blue line). Apart from the major displacements in the eastern wall, other minor displacements in the south wall seem related to debris slides of collapsed terraces.

## 6. Conclusions

We aim to promote the use of radar satellites, especially Sentinel-1, to improve the monitoring capacity of mining stakeholders. In this study, we demonstrate the potential of using the Sentinel-1 constellation to complement the ground motion monitoring system in the Riotinto mine. After processing both ascending and descending C-band radar images for the period 2015–2017, we successfully integrated them via projection to a common reference system. The proposed integration method managed to provide a joint dataset with complementary gaps filled and consistent measurements in intersection areas. Four main deformation zones are mapped, two of them unknown before this analysis, and displacement mechanisms are proposed for each of them. The displacements detected in areas under surveillance are in a remarkable correlation with terrestrial observations. The major displacements in the Atalaya pit are interpreted as a block slide restricted to the fault zone. Velocities up to 11 cm/y are observed, in a good correlation with the local GNSS network. Minor debris slides are also observed related to collapsed terraces in direction N-S, which is consistent with the visual observation even though they occur in the direction of low sensitivity for Sentinel-1. These results, coupled with the autonomy to provide high-quality unbiased data repeatedly at low cost, show the outstanding potential of Sentinel-1 to complement and improve ground monitoring systems. As for the limitations to consider, Sentinel-1 measures displacements only in the line of sight (moving away or approaching the satellite), they have a spatial resolution in the limit of the needs of the mining sector, and they lose information due to signal decorrelation in terrains under intense transformation. Therefore, Sentinel-1 measurements need to be integrated with ground-based observations to build an efficient early warning system. We are currently working on the automation of processing tools to integrate Sentinel-1 into the Riotinto mine

control system and thus contribute to a safe and sustainable supply of mineral resources necessary for the technological transition.

**Author Contributions:** Conceptualization, J.E. and I.M.; methodology, I.M., J.E. and J.F.; software, J.E. and I.M.; validation, F.T. and A.F.; formal analysis, D.M. and R.C.; investigation, J.E. and I.M.; resources, A.F. and F.T.; data curation, D.M. and R.C.; writing—original draft preparation, I.M. and J.E.; writing—review and editing, D.M, F.T., A.F., R.C., M.S. and J.F.; visualization, I.M. and J.E.; supervision, J.F. and M.S.; project administration, J.F. and M.S.; funding acquisition, J.F. and M.S. All authors have read and agreed to the published version of the manuscript.

**Funding:** This research was mainly funded by Copernicus (EC DG-GROW) and EIT RawMaterials through the RawMatCop program, grant number 248/G/GRO/COPE/16/9007 and 271/G/GRO/COPE/17/10036. Another important source of funding was the Spanish Agencia Estatal de Investigación research project DEEP-MAPS, grant agreement number RTI2018-093874-B-I00.

**Data Availability Statement:** Copernicus Sentinel data [2015,2016,2017], retrieved from Copernicus Open Access Hub, and processed by ESA.

**Acknowledgments:** This work has been supported by Atalaya Mining facilitating access and logistics to the mine, as well as providing the necessary data to complete the study. On the other hand, we sincerely thank all the members of the RawMatCop team for their support and enriching contributions. We would like to thank DARES S.L. for providing us with the A-DInSAR processing software. M.S. thanks the Ministerio de Ciencia, Innovacion y Universidades (project SANIMS, RTI2018-095594-B-I00). We thank the comments by Guillermo Marshall in a previous version of this manuscript.

**Conflicts of Interest:** The authors declare no conflict of interest.

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
