# Peer review of "Radar Interferometry as a Monitoring Tool for an Active Mining Area Using Sentinel-1 C-Band Data, Case Study of Riotinto Mine"

_remotesensing, doi:10.3390/rs14133061_

Round 1
Reviewer 1 Report
Abstract
Page 1 - Line 19: “… with improved resolution” Can the authors quantify the improvement in resolution?
Introduction
Page 2 - Line 61: The authors should consider and mention the anomalies currently suffering sentienl-1B.
Page 2 - Line 64-65: The authors must consider that there are companies that offer these services in the mining sector, therefore this technology is widely used in the mining sector.
InSAR historical review (ERS, ENVISAT, Sentinel-1)
Page 4 - Line 154: It is suggested to note that the three datasets work in the C band of the electromagnetic spectrum.
Page 4 - from Line 163: It is suggested to briefly describe the CPT method and include a flowchart of the interferometric processing chain.
Results and Discussion
Page 13 - Line 409: Can the authors justify why they have used only GNSS data in a single area to validate the DINSAR results?
Conclusions
Page 14 - Line 447: It is suggested to add that the radar images corresponded to Sentinel-1 (C-band)
Page 14 - Line 449: can the authors quantify the improvement in the obtained resolution?
Page 14 - Line 456: can the authors quantify this correlation?
Page 14 - It is suggested included limitations and shortcomings, and define future lines of work.
Author Response
Response to Reviewer-1 comments
We greatly appreciate the reviewer's in-depth review and pertinent comments that have improved our manuscript. We have replied to all the comments and modified the text according to the suggestions made. Please see the attachment.

Reviewer 2 Report
It is important in the future to take into account the isotopy of the information (any information positioned in the same pixel centers) and the information support, different in case of ground data (point support) and pixel (resolution area or regularised support). This will call for improving the spatial correlation analysis (already used empirically in case of outliers identification) and for considering a probabilistic approach, almost for studying the correlation among different data.
Author Response
Response to Reviewer-2 comment
We greatly appreciate the review of our work and the comment made. The reviewer shows us a future line of work demonstrating a deep understanding of the challenge of integrating radar data with different geometry. Certainly, although we observe a good spatial correlation in the integration of both geometries, the superposition of pixels from different projections is not something trivial. In many cases the pixel overlap will be partial, in addition, the size of the geocoded pixels will be different for each geometry since the angles of incidence with the topography will almost always be different. This opens a future line of work for a deeper analysis of what happens at the pixel level in the integration of data from different geometries.